# Impact of Incorporating Free Calcium and Magnesium on the Heat Stability of a Dairy- and Soy-Protein-Containing Model Emulsion

**DOI:** 10.3390/polym15224424

**Published:** 2023-11-16

**Authors:** Wei Wang, Kevin Wei Jie Tan, Poh Leong Chiang, Wai Xin Wong, Wenpu Chen, Qi Lin

**Affiliations:** Abbott Nutrition Research & Development (ANRD), Singapore 20 Biopolis Way, #09-01 Centros Building, Singapore 138668, Singapore; wang.wei1@abbott.com (W.W.); weijiekevin.tan@abbott.com (K.W.J.T.); pohleong.chiang@abbott.com (P.L.C.); waixin.wong@abbott.com (W.X.W.)

**Keywords:** milk protein concentrate, soy protein isolate, gelation, calcium, magnesium, viscosity

## Abstract

This study investigated the impact of calcium chloride (CaCl_2_) and magnesium chloride (MgCl_2_) at varying concentrations on a model milk formulation’s physical and chemical properties after thermal treatment. The model milk was subjected to two-stage homogenization and pasteurization before being supplemented with different concentrations of CaCl_2_ or MgCl_2_. The findings revealed that elevating the concentration of either calcium or magnesium resulted in the milk emulsion having a higher viscosity and median particle size following heating. CaCl_2_ had a slightly stronger impact than MgCl_2_, particularly at higher concentrations. The milk samples also exhibited a reduction in the zeta potential as the ionic strength of the salt solution increased, with the CaCl_2_-fortified milk displaying a slightly lower negative surface charge than the MgCl_2_-fortified milk at the same dose. The model milk’s viscosity was evaluated after adding various salt concentrations and a temperature ramp from 20 to 80 °C. Notably, the viscosity and particle size changes demonstrated a non-linear relationship with increasing mineral levels, where a significant increase was observed at or above 5.0 mM. An emulsion stability analysis also revealed that the de-stabilization pattern of the high salt concentration sample differed significantly from its low salt concentration counterparts. These findings could serve as a basis for the future development of fortified UHT milk with nutritionally beneficial calcium and magnesium in industrial applications.

## 1. Introduction

The demand for ultra-high-temperature (UHT) ready-to-drink (RTD) milk with a high protein concentration has been proliferating in the nutritional milk product category, especially during and after the COVID-19 era, as consumers are more health-conscious and aware of the benefits of immunity-boosting products [1]. The increasing demand for high-protein beverages reflects a trend in which consumers take a more proactive approach to health by incorporating nutrient-rich foods instead of relying solely on pharmaceuticals [2]. Additionally, consumers also seek cost-effective ways to supplement functional nutrients and achieve health benefits [3]. For example, the intake of adequate calcium may benefit older individuals’ skeletal systems with dietary protein, which promotes the production of insulin-like growth factor-1 (IGF-1), a substance that stimulates osteoblast-mediated bone formation [4]. Similarly, magnesium is crucial in preserving tissue integrity, supporting cellular functions, and enhancing the immune system [5]. From the perspective of human nutritional science, the low natural content of magnesium (110 mg/L; a serving of 250 mL contains <10% of the recommended dietary intake for adults) in UHT-processed milk makes it suitable for fortification [6]. Therefore, magnesium fortification can be widespread in nutritional dairy products.

However, achieving the thermal stability of high-protein UHT milk during the heat processing stage or throughout its shelf life can present a significant challenge. The high protein concentration in UHT milk can reduce the heat stability of the milk by reducing the heat coagulation time (HCT) through increased binding opportunities between protein micelles. The protein interaction can be further accelerated with free divalent cations, such as calcium and magnesium ions, by neutralizing the protein surface charge and reducing their particle repulsion forces [7]. Calcium ions can trigger milk gelation, where a higher concentration results in a shorter gelation time and a higher yield strength of the gels [8]. Increasing the fortification of calcium and magnesium in the high-protein UHT milk may reduce the thermal stability of the milk protein and cause gelation or fouling in the processing and sedimentation in the shelf life [9]. The thermally induced dissociation of ĸ-casein and the presence of calcium ions could also cause more undesired interactions between β-lactoglobulin and caseins [10]. Adding calcium chloride to milk might increase the micellar calcium and the hydrogen ion concentration in the milk with a concomitant decrease in the pH, which could potentially induce milk gelation [11,12]. Similarly, magnesium ions in milk can enhance the interaction with casein, forming a gel or coagulum at temperatures exceeding 70 °C, comparable to that of calcium ions [6]. However, free magnesium is sparsely studied compared with calcium on the heat stability of protein in milk. Previously, there were comprehensive studies on calcium-fortified milk’s heat stability, sediment formation, and rheological properties [13,14,15]. Studies on the impact of different minerals emphasized much more calcium fortification than magnesium [16,17]. As a result, the influence of magnesium on the rheological and physicochemical properties of UHT milk could be underestimated without appropriate control, which could lead to processing issues such as fouling in the heat exchangers or product quality issues in the shelf life due to protein aggregation and coagulation. Additionally, several authors reported differences in the association behavior of calcium and magnesium with caseins [18,19]. Free calcium and magnesium exhibit variances in the water hydration levels and their propensities to bind with protein and casein. The less hydrated ions possess greater capacities to induce the participation of sodium caseinate in water after heating [20]. The low percentage of magnesium association with casein micelles can be explained by the low saturation indexes of salts like MgHPO_4_ and Mg_3_(PO_4_)_2_ [18]. Although the study proposed that the added magnesium might interact with casein micelles less than calcium, the same work found that the addition of 8 mM of magnesium ions resulted in casein precipitation after heat treatment instead of calcium ions [18]. 

Recently, the consumption of soy-based milk has increased because of the absence of cholesterol and lactose, and it is becoming an extension of milk protein in food systems for a population with lactose intolerance and heart disease [21,22]. Soy protein isolate (SPI) primarily consists of two subunits, glycinin (11S) and β-conglycinin (7S), which collectively account for 70% of the total storage proteins present in soybean seeds [23,24]. However, low concentrations of magnesium ions facilitate soy protein aggregates by boosting the calcium ion coagulation power [25]. There is a need for more research on the combination of milk protein and soy protein in the high-protein UHT beverage and the impact of calcium and magnesium on its thermal stability and shelf-life stability.

This study examined the influence of magnesium incorporation compared to calcium on various physical properties of a model UHT milk with dairy and soy proteins at elevated temperatures. The milk model (comprising a blend of milk and soy proteins) was first produced via homogenization and High-Temperature Short Time (HTST) thermal treatment for protein hydration was used as a general practice for UHT milk in the industry. Subsequently, the milk sample was treated with varying levels of calcium or magnesium via aseptic dosing and heated again to investigate the impact of the mineral dosage level on its physicochemical properties pre and post heat treatment. The objective was to gather additional information about the degree to which milk stability could be compromised through calcium or magnesium enrichment and to establish a correlation between the physicochemical properties of the milk samples before the heat treatment, the rheological behavior during the heat treatment, and the dispersion stability of the emulsion system after the heat treatment.

## 2. Materials and Methods

### 2.1. Materials

Soy protein isolates (SPIs) was obtained from Solae (International Flavour & Fragrances, Singapore), milk protein concentrate (MPC 70) was obtained from Fonterra (Auckland, New Zealand), soy oil was obtained from the local vendor (Fuji oil, Osaka, Japan), and maltodextrin with dextrose equivalent (DE) 25 was obtained from Cargill (Shanghai, China). Calcium chloride and magnesium chloride were purchased from Sigma Aldrich (Singapore). All other reagents used were of analytical grade.

### 2.2. Preparation of the Model Milk

The model milk with 5% soybean oil, 20% maltodextrin, and 6.3% protein with MPC and SPI (1:1) combination was prepared at a batch size of 5 kg. A pilot-scale UHT system (OMVE, Utrecht, The Netherlands) was used to pasteurize the milk at 80 °C for 30 s after high shear mixing (pre-emulsification) and two-stage high-pressure homogenization at 250/50 bar. The sample was filled into bottles aseptically and stored in the refrigerator for further treatment. CaCl_2_ and MgCl_2_ stock solutions were prepared at 1M concentration and added to the model milk to reach concentrations of 0, 2, 3.5, 5, and 6.5 mM calcium and magnesium, respectively, before sample analysis.

### 2.3. Conductivity

Conductivity measurements were performed with a Mettler Toledo S230 SevenCompact conductivity meter (Greifensee, Switzerland) equipped with an Inlab^R^ 731 probe (Mettler Toledo, Greifensee, Switzerland).

### 2.4. Zeta Potential

The electrophoretic mobilities, and hence, the calculated zeta potentials, were determined via electrophoresis and phase analysis light scattering (PALS) using a NanoBrook Omni zeta potential analyzer (Brookhaven Instruments Co., Holtsville, NY, USA). Particle Solutions version 3.6 (Brookhaven Instruments) was used to collect and analyze data. After dilution, the samples were measured in triplicate (repeating analysis) in 10 mm polystyrene cuvettes. The zeta potential was calculated from the electrophoretic mobility using the Smoluchowski equation [26].

### 2.5. Rheological Analysis

Temperature ramps were conducted using ARES G-2 strain-controlled rheometer from TA Instruments (TA Instruments, Newcastle, DE, USA). A 32 mm concentric cylinder geometry was used with a constant shear rate setting at 100/s. An aliquot of a 5 mL sample was added into the cup each time in the measurement, and a vapor trap was applied during the temperature ramp to minimize water evaporation. Measurement started at 20 °C and ramped up to 80 °C at a rate of 3 °C/min. After 10 s holding at 80 °C, the temperature was ramped down to 20 °C at a similar rate. The starting, ending, and minimum viscosities measured during the temperature program were collected to compare the viscosity change among different samples. The sample’s critical temperature (Tc), which marks the point at which protein aggregation would occur, was identified as the temperature at which the minimum viscosity was measured.

### 2.6. Particle Size Distribution Measurement

Particle size distribution of the liquid sample was analyzed using the Malvern Panalytical Mastersizer 3000 (Malvern Instruments Ltd., Malvern, UK) with the Hydro LV liquid module. A 5–15% obscuration was selected for sample loading. The refractive index used for this analysis was 1.47.

### 2.7. Dispersion Stability Analysis

Accelerated emulsion stability was analyzed using LUMisizer (L.U.M. GmbH, Berlin, Germany). An amount of 460 µL of sample was loaded into the 2 mm measurement cell and centrifuged at 1800× *g* for 1 h at 20 °C. The instability index was analyzed based on the change in near-infrared (NIR) transmission during the analysis. Sediment height was also determined at the end of the analysis. Values are presented as mean ± standard deviation of data from three independent trials.

### 2.8. Statistical Analysis

One-way analysis of variance (ANOVA) was conducted for the determination of differences between samples using the SPSS 27.0 software (IBM, White Plains, NY, USA). Duncan’s test was employed. A probability level of *p* ≤ 0.05 was considered to be significant for all statistical procedures.

## 3. Results and Discussion

### 3.1. Interaction between Added Divalent Cations and Proteins before Heat-Treatment

The average calcium and magnesium contents in a standard 100 g serving of UHT milk were approximately 117 mg and 11 mg, respectively [27]. However, these levels are not sufficient to meet the daily requirements for adults, with one 330 mL serving falling short of the recommended intake levels of calcium (1200 mg/day) and magnesium (320 mg/day) [28]. Alternatively, calcium and magnesium fortification are the best ways to enhance the nutritional value of dairy milk, which prompts an understanding of the possible impact of the soluble calcium or magnesium added to the milk. The pH, conductivity, and zeta potential measurements were conducted to explore their correlations and potential impacts on milk stability due to mineral fortification with various dosages and types to understand the change in milk concerning the chemical and physical properties before the heat treatment. 

Table 1 indicates that increasing the CaCl_2_ and MgCl_2_ addition can lead to a decrease in the pH. Meza et al. [14] reported a similar trend in the calcium-added reconstituted skim milk solution. The addition of calcium ions can alter the original ion thermodynamic equilibria and promote calcium phosphate formation by deprotonating H_2_PO_4_^−^ (7.5 mmol/L at pH = 6.7) in the milk through an interaction with HPO_4_^2−^ to form amorphous CaHPO_4_·2H_2_O, which would be changed into its crystalline form over time [29]. Similarly, adding magnesium ions to the milk beverage may form insoluble tri-magnesium phosphate (solubility product K_SP_: 10^−24^) [29]. The release of additional H^+^ ions during deprotonation causes a decrease in the pH.

The conductivity value of milk can be measured as a parameter for milk quality appreciation, where the electrical conductivity of milk could increase during mastitis due to an increase in Na^+^ and Cl^−^ [30]. In this case, the addition of calcium and magnesium salts led to increased conductivity due to the increase in the concentrations of cations and anions at the natural pH of milk (Table 1). A linear relationship was observed between the measured conductivity and the ionic strength of the serum phase in the milk protein concentrates [31]. Therefore, conductivity was used to estimate the ionic strength change in the serum phase before and after mineral fortification in the model system. In this study, the conductivity of milk with added calcium or magnesium increased proportionally with the amount of salt added. Adding 2 mM of calcium or magnesium divalent salts to the milk model still increased the samples’ overall conductivity significantly (Table 1), which suggests that the buffering capacity of the model milk samples against conductivity change is feeble, and that the addition of CaCl_2_ or MgCl_2_ contributes free divalent ions in the serum phase of the milk model. The conductivity test results indicating the milk solutions’ ionic strengths also do not show statistically significant differences between the free calcium and magnesium-fortified milk samples at equal mineral dose levels (Table 1).

The importance of κ-casein in maintaining casein micelle stability has been recognized for a long time. The association between casein and micelles is believed to occur in a manner where the negatively charged hairy peptide, especially phosphoryl clusters, extends into the water. This extension repels casein micelles, preventing them from binding to each other. Both electrostatic repulsion and steric hindrance work together to prevent the caseins from polymerizing and growing [32]. An increased ionic strength in the milk solution will weaken the electrostatic repulsions of casein micelles and lead to protein aggregation at lower temperatures. The measured zeta potential for the control sample without CaCl_2_ and MgCl_2_ addition was −39.5 mV, close to the result reported for the fat globules in reconstituted milk [33]. The measured zeta potential values of the other samples with the addition of variable amounts of CaCl_2_ or MgCl_2_ were below −30 mV, indicating changes in the fat globules’ surface charge, which may lead to emulsion degradation [34,35]. The zeta potentials in the soy-protein-stabilized emulsion with increasing doses of CaCl_2_ or MgCl_2_ had a similar decreasing trend [7]. The calcium-fortified milk also had a slightly weaker zeta potential value than its magnesium-fortified counterparts (Table 1). Since calcium ions are more tightly bound to the casein micelle coating on the fat globular surface in milk, the effect of calcium ions on the surface charge reduction was more significant than that of magnesium. The result agrees with the research findings on casein micelles and soy proteins [18,36]. A modified Jones–Dole equation was previously used in another study to determine the hydration coefficients of various alkaline earth metal ions in water, including calcium and magnesium. The study found that calcium cations exhibited a lower degree of hydration in water than magnesium, which likely contributed to their higher capacity to induce the precipitation of sodium caseinate [20]. The reduction in the zeta potential of milk and soy protein may de-stabilize the solution due to reduced electrostatic repulsion between the particles dispersed in water until the repulsion force becomes weak enough to cause subsequent aggregation or flocculation. Hence, adding CaCl_2_ or MgCl_2_ to the milk model may cause thermal instability and aggregation. 

### 3.2. Rheological Properties of the Heat-Treated Protein Solutions with Added Divalent Cations

The impact of the addition of calcium and magnesium ions on the milk and soy protein mixed system was studied by monitoring the changes in the apparent viscosity as a function of temperature using a rheological method. Thermal treatments from 20 °C to 80 °C were applied to all samples to see the milk viscosity change with variable concentrations of divalent cations. Figure 1 shows that the viscosity of all samples exhibit a similar trend. For all samples except for the control, the liquid viscosity decreased with the increasing temperature until it reached a critical temperature (Tc). The viscosity value corresponding to the Tc was identified to be the minimum viscosity, where the temperature-dependent viscosity decrease went to the end. In this stage, the interaction energy of the colloidal forces concerning the Brownian thermal energy decreased as the temperature increased, without protein aggregation. A similar trend was observed for the viscosity of a milk protein concentrate dispersion when the temperature increased from 25 °C to 70 °C in the heat treatment [37]. The denaturation of α-lactalbumin and β-lactoglobulin induces whey protein aggregation via disulfide bridging, whereas the interaction between whey protein and casein does not occur because caseins do not undergo thermal denaturation in this temperature range [38]. Similarly, when the temperature is above 70 °C, β-conglycinin from soy protein starts to dissociate into subunits with structural changes and forms soluble complexes between the β-subunit of β-conglycinin and the basic subunit of glycinin, which could also lead to the increase in viscosity [39,40]. However, it has been suggested that the thermal treatment did not cause a chemical interaction between the soy proteins and casein micelles in the mixed system [41]. Conversely, another study suggested that the whey protein and soy protein might interact to form aggregates under a thermal treatment via disulfide bridges [42]. In this study, the potential interaction between the soy and milk protein could not be ruled out because of the observed viscosity change after the heat treatment, particularly at higher levels of mineral fortification.

The increase in the viscosity with the increasing temperature was observed when the temperature was above 75 °C. As shown in Table 2, the Tc of the milk samples decreased with the increasing calcium or magnesium addition. MgCl_2_ was slightly less effective than CaCl_2_ in reducing the Tc during the heat treatment, especially in the samples with 5.0 mM and 6.5 mM of salt addition. One possible explanation is that those divalent salts could create crosslinkages between casein micelles, resulting in a higher apparent viscosity and reduced shear-thinning properties compared to the control [43]. The interiors of casein micelles are mainly composed of α (S_1,2_)- and β-caseins, while κ-casein is located on their surfaces, acting as a block copolymer with a non-adsorbed block extending into the solution, forming a hairy layer that prevents neighboring micelles from merging. The addition of calcium resulted in the loss of the particle-stabilizing properties of the hairy layer, leading to the attraction between adjacent micelles and changing their colloidal properties [44]. Another possible reason was that the decreasing pH of the model milk due to the release of hydrogen ions accelerated the association of casein micelles [44]. In soy proteins, calcium ions can bind to the carboxyl group of the glutamyl and asparagine residues in the 11S subunit in the neutral pH region (pH 6.0–8.0) [36,45]. An increased calcium content (from 2.87 mg/g protein) can lead to the aggregation of soy protein due to exothermal aggregation and the disruption of the hydrophobic interaction that contributes to stabilizing aggregates promoted by the ions [46]. The formation of MgCl_2_-induced gels for SPI is primarily due to the ionic attractions between the protein molecules and magnesium ions, although the gelation of SPI and 11S is also affected by other molecular forces, such as hydrophobic interactions and hydrogen bonds [47]. 

The temperature above the Tc changed the viscosity depending on the mineral dosage and type. The milk samples that were fortified with higher mineral concentrations exhibited more rapid viscosity evolutions than those with lower salt concentrations (Figure 1). After undergoing the reversed cooling process, viscosity measurements were performed for all of the samples, and Table 2 shows a substantial variation in the final viscosity of each sample. The thixotropic effect acted as the dominant force for the low-salt-concentration samples, with little protein aggregation occurring. Conversely, the samples that were treated with 5.0 mM and 6.5 mM of calcium or magnesium exhibited considerably higher viscosities, indicating irreversible changes such as protein aggregation in the milk system. When comparing the effect of magnesium to calcium on the viscosity after the heat treatment, it was found that magnesium had a comparable effect to calcium but was slightly less pronounced. This observation is consistent with the results obtained from the zeta potential measurement, as presented in Table 1.

### 3.3. Particle Size Analyses

Figure 2a,b show the emulsion’s particle size distribution and mean particle diameter with different concentrations of CaCl_2_ and MgCl_2_. These measurements were carried out after the salt addition and thermal treatment. The emulsion without CaCl_2_ and MgCl_2_ displays a bimodal particle size distribution with a median (D_50_) particle diameter of 1.7 µm after the heat treatment. This bimodality may arise due to the dual system of oil droplets coated by protein molecules on surfaces and small protein aggregates suspended after homogenization in the same system. Adding CaCl_2_ or MgCl_2_ at 2.0 mM or 3.5 mM did not significantly alter the mean diameter, indicating that low concentrations of CaCl_2_ or MgCl_2_ (below 3.5 mM) cannot induce fat globule aggregation. However, at 5.0 mM CaCl_2_ or MgCl_2_, there was a noticeable increase in the mean diameter. The particle size distribution shifted towards a larger range, suggesting that extensive aggregation occurred in the original emulsion droplets. Furthermore, an increase in CaCl_2_ or MgCl_2_ addition to 6.5 mM led to more aggregation of the proteins and flocculation of the fat droplets. More calcium or magnesium addition would be anticipated to reduce the energy barrier required, potentially accounting for the sudden emergence of protein aggregation in the samples containing 5.0 mM and 6.5 mM mineral levels, with the temperature being high enough to prompt protein aggregation. Based on the findings above, it can be concluded that a concentration of 3.5 mM of CaCl_2_ or MgCl_2_ is still within the maximum amount of salt that is allowable to prevent extensive droplet aggregation and thus still gives a relatively stable milk model emulsion. This finding aligns with previous studies, suggesting that adding calcium could cause droplet aggregation in whey-protein-stabilized emulsions [48,49]. The increase in droplet aggregation was likely due to fat globule flocculation rather than coalescence [48]. This was mainly caused by reduced electrostatic repulsion between oil droplets stabilized by milk or soy protein upon adding CaCl_2_ or MgCl_2_.

The fact that low concentrations of CaCl_2_ or MgCl_2_ did not induce flocculation suggests that the emulsion has a specific capacity to withstand the reduction in electrostatic repulsion. Additionally, two modes were detected in the particle size distribution for the samples before and after the rheological measurement. The primary mode presented a particle size in the nano range, attributed to the oil droplets following homogenization. The secondary mode exhibited a broader range of particle sizes and was suggested to correspond to protein particles, as confirmed via a microscopic examination (Appendix A). In this investigation, protein aggregation was insignificant at 2.0 mM and 3.5 mM salt concentrations but markedly pronounced at higher mineral levels. This was most likely caused by the pH and ionic strength change in the emulsion at higher salt concentrations. The aggregation of food proteins in an aqueous dispersion depended on the effects of protein concentration, pH, and ionic strength [50]. The pH and ionic strength can also alter the shapes of the protein aggregates [51]. An investigation of the thermal aggregate of calcium-enriched milk also discovered that protein aggregation resulted in disordered aggregates that occluded a significant quantity of solvent [14]. This study observed a similar shape of protein aggregates in the emulsion despite the addition of calcium or magnesium salts. 

### 3.4. Emulsion Stability

The LUMisizer, a measurement instrument employing STEP Technology, enables the simultaneous assessment of the intensity of transmitted light as a function of time and position over the entire sample length. The physical stability of the milk model with added CaCl_2_ or MgCl_2_ at variable levels was analyzed using the LUMisizer (Figure 3). As per the methodology, the integrated transmission–time profiles were used to indicate the dispersion stability. The transmission profiles reflect the variation in droplet concentration inside the emulsion and the sediment at the bottom. A high transmission indicates a low concentration of droplets, whereas a low transmission indicates a high concentration of droplets, thus suggesting aggregation or coalescence. Emulsions that undergo a more significant transmission change during centrifugation are less stable [52].

The uniform distribution of the emulsions with variable levels of Ca and Mg ions at the start of the test is evident (refer to Figure 3 and Figure 4), and the initial profile line is predominantly linear in shape. However, with an increased centrifugation time, the transmittance curves for the emulsion samples gradually exhibited differences in profiles. The time evolution resulted in varying separation behaviors for the emulsions with different levels of added CaCl_2_ or MgCl_2_. The rapid increase in transmittance at the bottom of the sample was accompanied by a slow increase in the upper part of the sample cell instead, suggesting that the oil droplets in the emulsion were migrating upwards. Over time, a sedimentation layer occupying the bottom of the sample can be observed, represented by the shadowed region with a low transmission rate on the right side of the transmission profiles (Figure 3 and Figure 4). This finding was consistent with previous studies investigating the emulsion stability of oil droplets coated with protein [53,54].

With increasing amounts of calcium or magnesium addition, the transmission profiles of the different samples appeared to be similar until 5.0 mM of calcium or magnesium addition was reached. In this stage, the samples exhibited a typical clarification process for polydispersed particles in colloidal suspension during the LUMisizer analysis, which was supported by the particle size distribution results (Figure 2) and the viscosity measurement (Table 2) shown earlier, where the viscosity and particle size distribution of the samples with mineral fortification remained similar to the control. Above 5.0 mM of mineral fortification, the viscosity of the milk samples increased significantly, suggesting that there was inter-particle crosslinking between the soluble part of the proteins after the heat treatment. Simultaneously, it is speculated that the insoluble portions of soy protein isolates and the casein micelles entangle and bind together with the soluble proteins through crosslinking with disulfur bonds or sulfhydryl groups. Therefore, separating the insoluble proteins becomes harder, with less sample separation observed in the LUMisizer results for the samples with 5.0 mM mineral fortification. 

Nevertheless, the emulsion became highly unstable after the mineral level reached 6.5 mM, as evidenced by a significant increase in the transmission over the test duration (as shown in Figure 3e and Figure 4e). With an increased mineral concentration, the protein aggregation became more severe by crosslinking through disulfide bonds and sulfhydryl groups, resulting in the formation of large protein aggregates, as measured and represented by the second prominent peak in the particle size distribution curves with 6.5 mM CaCl_2_ or MgCl_2_ (Figure 2). Consequently, large aggregates with oil droplets flocculated should have lower densities than that of water and therefore migrate upwards to accumulate at the top of the liquid sample after centrifugation. Thus, the transmission profile of the sample at 6.5 mM minerals was dominated by the migration of large protein aggregates instead of the suspended colloidal particles, with calcium-fortified milk showing a higher transmission change and worse stability than magnesium.

The stability of the emulsion can be determined by measuring the instability index, which is proportional to the degree of separation caused by particle sedimentation or the creaming of oil droplets [55,56]. Although the LUMisizer analysis based on centrifugation is commonly employed to assess the dispersion stability of liquid samples, it is based on Stokes’ law. It is usually limited to samples without a gel network. The instability index results of various test samples are presented in Table 3. Because of the protein crosslinking and network formation during the heat treatment, the instability index measured for the samples with a high mineral fortification may not represent their actual emulsion stabilities. The sedimentation heights of all of the samples with a variable amount of calcium or magnesium are presented in Figure 3 and Figure 4, with the calculated values shown in Table 3. The sample solution varies in sediment height, as evidenced by the development of an optically dense (low transmission) area at the bottom of the sample cell. For the samples without protein aggregation, we can expect more sedimentation from the protein particles compared to large protein aggregates with flocculated oil droplets. The variability of the samples’ instability indexes and sedimentation heights between samples with calcium versus magnesium at 5.0 mM and 6.5 mM indicates that the protein aggregation and the development of a gel network with calcium addition are more robust than those caused by magnesium, which also agrees with the particle size distribution results and the viscosity measurements shown earlier.

## 4. Conclusions

This study investigated the impact of magnesium and calcium fortification on the physicochemical properties of UHT model milk. The results showed that the addition of either magnesium or calcium led to a decrease in the surface charges of the protein particles and a slight reduction in the sample pH. However, the impact of MgCl_2_ on reducing the zeta potential of the protein particles was found to be lower than that of CaCl_2_, which could explain the observed differences in the particle size and viscosity between the samples treated with the same concentrations of each mineral. When the heat-treated samples contained more than 5.0 mM of added salt, there was a marked alteration in the viscosity and particle size, suggesting that it could be the upper limit of tolerance of the model milk to salt. For further studies, this could be a useful gauge to investigate other ways to increase salt concentrations beyond 5.0 mM with varying protein systems according to manufacturing needs. Although the instability index based on Stokes’ law was not highly correlated with the actual condition due to the formation of a protein gel network, the transmission profiles and sedimentation heights from the LUMisizer analysis provided a more meaningful interpretation of the results.

## Figures and Tables

**Figure 1 polymers-15-04424-f001:**
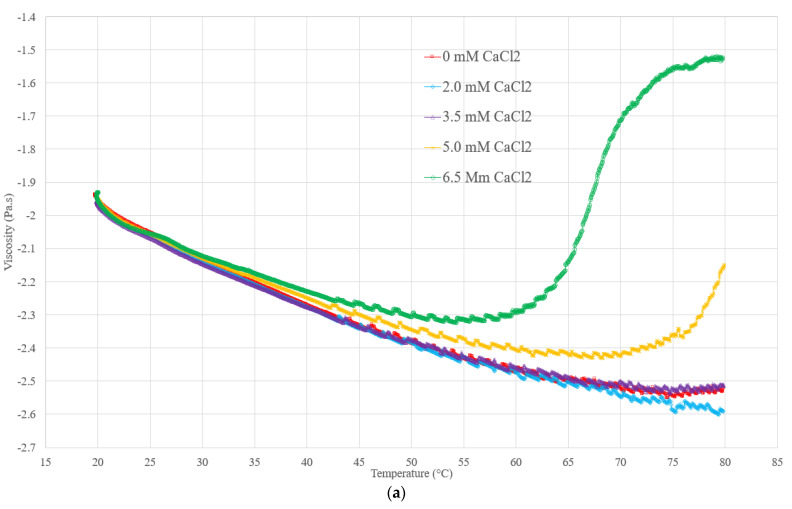
(**a**,**b**) show the viscosity profile of samples with different salt additions (calcium and magnesium), with temperature ramping to reach 80 °C by using the liquid rheometer.

**Figure 2 polymers-15-04424-f002:**
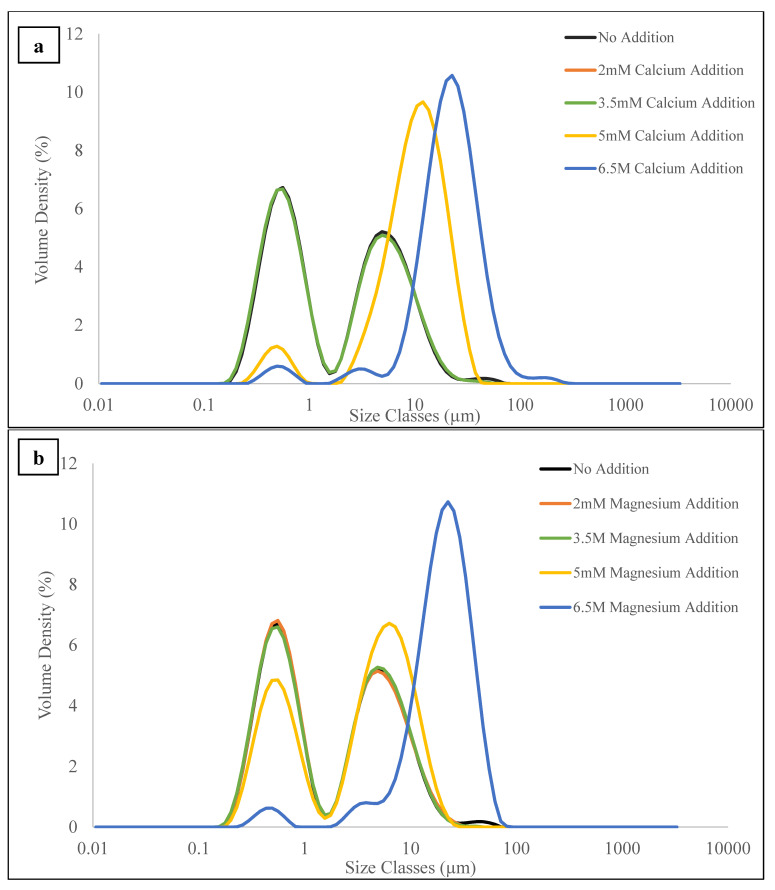
Particle size distribution curves of samples with (**a**) 0, 2.0, 3.5, 5.0, and 6.5 mM of CaCl_2_ addition and (**b**) 0, 2.0, 3.5, 5.0, and 6.5 mM of MgCl_2_ addition.

**Figure 3 polymers-15-04424-f003:**
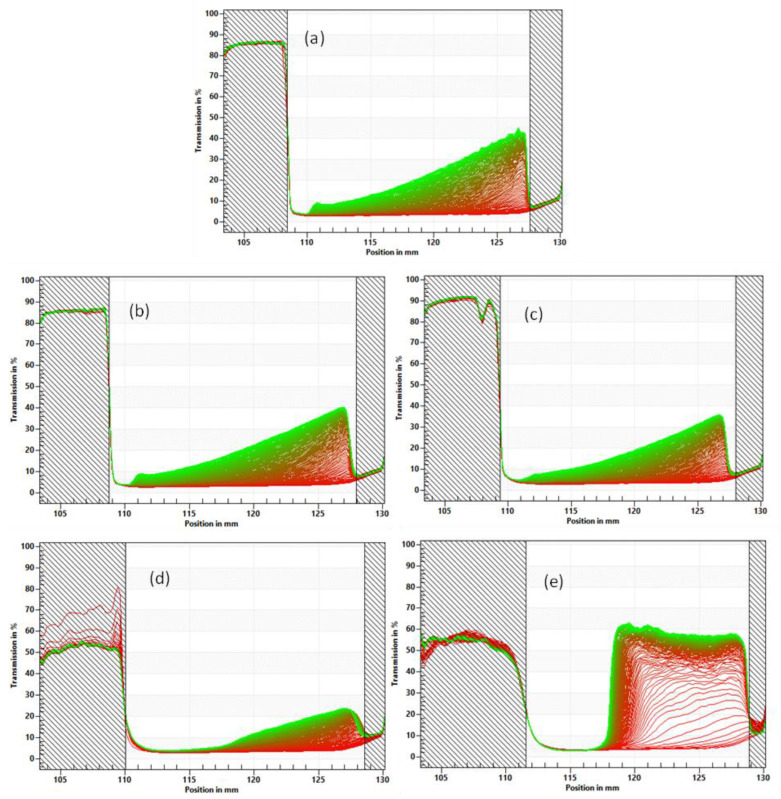
Transmission profiles of samples with different doses and sources of salt addition obtained via LUMisizer measurement; (**a**) 0 mM, (**b**) 2.0 mM CaCl_2_, (**c**) 3.5 mM CaCl_2_, (**d**) 5.0 mM CaCl_2_, (**e**) 6.5 mM CaCl_2_.

**Figure 4 polymers-15-04424-f004:**
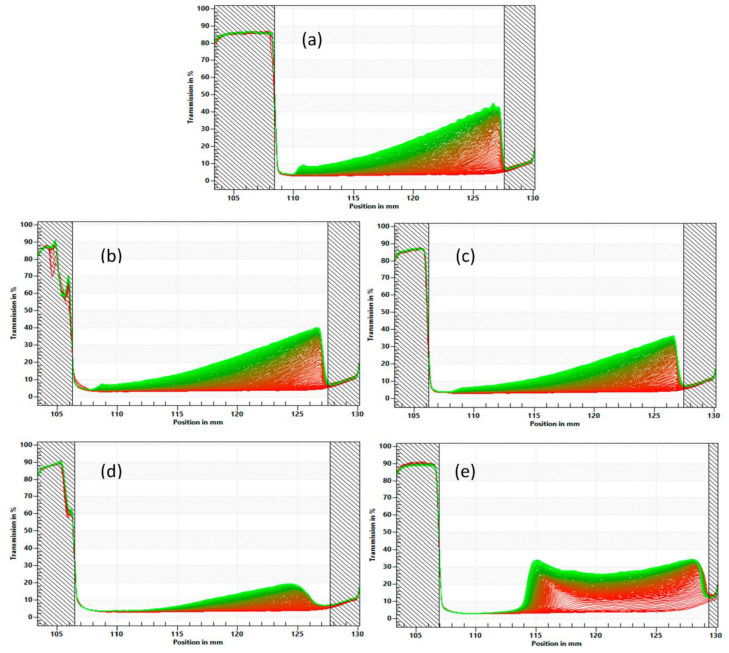
Transmission profiles of samples with different doses and sources of salt addition obtained via LUMisizer measurement; (**a**) 0 mM (**b**) 2.0 mM MgCl_2_ (**c**) 3.5 mM MgCl_2_ (**d**) 5.0 mM MgCl_2_ (**e**) 6.5 mM MgCl_2_.

**Table 1 polymers-15-04424-t001:** Values of pH, conductivity, and zeta potential of the milks with different levels of MgCl_2_/CaCl_2_ *.

	Mineral Dose Level	pH	Conductivity of Milk (µS/cm)	Ζeta Potential (mV)
No addition	Zero	6.93 ± 0.03 ^e^	1072 ± 29 ^a^	−39.5 ± 2.6 ^a^
CaCl_2_ addition	2 mM Ca	6.86 ± 0.03 ^d^	1185 ± 33 ^b^	−14.6 ± 0.3 ^cd^
	3.5 mM Ca	6.81 ± 0.05 ^b^	1267 ± 23 ^c^	−12.3 ± 0.7 ^de^
	5 mM Ca	6.76 ± 0.05 ^ab^	1349 ± 15 ^d^	−9.9 ± 1.7 ^efg^
	6.5 mM Ca	6.73 ± 0.06 ^a^	1450 ± 19 ^e^	−8.9 ± 0.9 ^g^
MgCl_2_ addition	2 mM Mg	6.86 ± 0.03 ^d^	1186 ± 10 ^b^	−17.24 ± 1.1 ^b^
	3.5 mM Mg	6.84 ± 0.02 ^c^	1229 ± 19 ^bc^	−16.20 ± 1.6 ^bc^
	5 mM Mg	6.78 ± 0.01 ^ab^	1365 ± 25 ^d^	−11.60 ± 0.3 ^ef^
	6.5 mM Mg	6.75 ± 0.02 ^ab^	1465 ± 21 ^e^	−9.25 ± 0.0 ^fg^

* Different letters within each column indicating significant difference at *p* < 0.05.

**Table 2 polymers-15-04424-t002:** Viscosities of samples with/without mineral addition during temperature sweep (100/s) *.

	Mineral Dose Level	Starting Viscosity (mPa.s)	Minimum Viscosity (mPa.s)	Tc * (°C)	Ending Viscosity (mPa.s)
No addition	Zero	11.53 ± 0.15 ^a^	2.36 ± 0.03 ^a^	78.99 ± 0.68 ^f^	5.74 ± 1.26 ^a^
CaCl_2_ addition	2 mM Ca	13.43 ± 2.49 ^b^	2.51 ± 0.01 ^a^	79.59 ± 0.19 ^f^	9.71 ± 1.44 ^a^
	3.5 mM Ca	11.27 ± 0.13 ^a^	2.91 ± 0.08 ^b^	74.98 ± 0.34 ^e^	9.01 ± 0.29 ^a^
	5 mM Ca	11.11 ± 0.68 ^a^	3.71 ± 0.06 ^d^	66.18 ± 1.79 ^c^	31.86 ± 3.87 ^c^
	6.5 mM Ca	11.55 ± 0.21 ^a^	4.88 ± 0.21 ^f^	54.38 ± 0.56 ^a^	49.47 ± 5.74 ^e^
MgCl_2_ addition	2 mM Mg	10.76 ± 0.26 ^a^	2.54 ± 0.01 ^a^	79.37 ± 0.24 ^f^	6.43 ± 0.54 ^a^
	3.5 mM Mg	10.66 ± 0.24 ^a^	2.80 ± 0.08 ^b^	74.00 ± 1.90 ^e^	8.25 ± 0.16 ^a^
	5 mM Mg	10.81 ± 0.24 ^a^	3.35 ± 0.18 ^c^	70.08 ± 2.59 ^d^	16.00 ± 1.23 ^b^
	6.5 mM Mg	10.93 ± 0.19 ^a^	3.96 ± 0.09 ^e^	60.45 ± 1.84 ^b^	41.82 ± 5.20 ^d^

* Different letters within each column indicate significant difference at *p* < 0.05.

**Table 3 polymers-15-04424-t003:** Instability index and sedimentation height obtained via LUMiSizer measurement *.

	Mineral Dose Level	Instability Index	Sedimentation Height (mm)
No addition	Zero	0.174 ± 0.007 ^d^	2.57 ± 0.15 ^c^
CaCl_2_ addition	2.0 mM Ca	0.162 ± 0.010 ^cd^ *	2.53 ± 0.29 ^c^
	3.5 mM Ca	0.132 ± 0.012 ^b^	2.50 ± 0.17 ^c^
	5.0 mM Ca	0.078 ± 0.010 ^a^	1.57 ± 0.21 ^b^
	6.5 mM Ca	0.366 ± 0.010 ^f^	1.15 ± 0.13 ^a^
MgCl_2_ addition	2.0 mM Mg	0.156 ± 0.004 ^c^	2.57 ± 0.06 ^c^
	3.5 mM Mg	0.132 ± 0.003 ^b^	2.50 ± 0.10 ^c^
	5.0 mM Mg	0.065 ± 0.004 ^a^	2.40 ± 0.10 ^c^
	6.5 mM Mg	0.206 ± 0.014 ^e^	0.93 ± 0.06 ^a^

* Different letters within each column indicate significant difference at *p* < 0.05.

## Data Availability

Data are contained within the article and Appendix A.

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
