# Peer review of "Impact of Incorporating Free Calcium and Magnesium on the Heat Stability of a Dairy- and Soy-Protein-Containing Model Emulsion"

_polymers, 2023, doi:10.3390/polym15224424_

Round 1

Reviewer 1 Report

Comments and Suggestions for Authors

The novelty of this article is that it examines the influence of magnesium incorporation compared to calcium on various physical properties of a model UHT milk with dairy and soy protein at elevated temperatures. The article claims that this is the first study to investigate the combination of milk and soy protein in the high-protein UHT beverage and the impact of calcium and magnesium on its thermal and shelf-life stability. The authors study the properties of UHT milk by separately adding magnesium and calcium salts and compare the properties with each other. But there is no information about what happens when both salts are added. This would be very useful for further development.

The manuscript has many gaps, for example:
The abstract is missing in the manuscript!
The reference style is inconsistent with the journal template.
Some abbreviations are not defined.
It is unclear why so specific axes range is set to Figure 1b.
It is not possible to read the values and axes' names in the figure 3 .

Author Response

Thanks to the positive comment from reviewer #1 about the novelty of this study and the suggestion for future. We would like to address the other comments from reviewer #1 with the following actions.

  1. The abstract is missing in the manuscript!

To reviewers’ convenience, abstract was re-included in the end of this letter for his/her review (please see the attachment).

  1. The reference style is inconsistent with the journal template.

Reference style checked and corrected accordingly

  1. Some abbreviations are not defined.

Abbreviations have been re-checked and defined where they were used.

  1. It is unclear why so specific axes range is set to Figure 1b.

The Y-axes range in Figure 1b was re-adjusted to make it consistent with Figure 1a. Graphs have been re-plotted to make it more readable

  1. It is not possible to read the values and axes' names in the figure 3.

To make the values more readable in Figure 3, we would like to break the graphs previously in figure 3 to become two parts. Figure 3 and Figure 4 to represent the LUMisizer profiles of samples with CaCl2 (Figure 3) and MgCl2 (Figure 4), respectively in the new manuscript.

 .

Reviewer 2 Report

Comments and Suggestions for Authors

This paper systematically investigated the impact of magnesium and calcium fortification on the physicochemical properties of UHT model milk. The article was well-written and organized, and the authors clearly analyzed the properties, including pH, Zeta potential, conductivity, viscosity, particle size, and emulsion stability. This paper is a good fundamental study that can provide a guide to the addition of calcium and magnesium in the high-protein UHT beverage.

I have several minor changes as stated following:

1.     Page 6 “MgCl2 seems slightly less effective than CaCl2 in reducing Tc during heat treatment, especially in the samples with 5.0mM and 6.5mM salt addition.”  I suggest not using the word ”seem” but instead using a more solid statement since your data supports this finding.

2.     Page 8 “This bimodality may arise due to the dual system of milk and soy protein.”  However, the author stated on page 9 that “the primary mode presented a particle size in the nano range, attributed to the oil droplets following homogenization. The secondary mode exhibited a broader range of particle sizes and was suggested to correspond to protein particles, as confirmed by microscopic examination”.  From my point of view, the two explanations for this bimodality are contradictory. Can the authors provide an explanation?

3.     Page 8 “Based on the findings above, it can be concluded that a concentration of 3.5mM CaCl2 or MgCl2 is the maximum amount of salt allowed to prevent extensive droplet aggregation and thus still give a relatively stable milk model emulsion.” However, the author mentioned that MgCl2 is less effective than CaCl2, and thus, the max value should be different. I suggest the author use a more accurate description.

4.     Page 9 “In our study, we observed a similar shape of protein aggregates in the emulsion in spite of calcium or magnesium salts addition.” Images should be provided if you would like to discuss these results.

Author Response

Reviewer #2

This paper systematically investigated the impact of magnesium and calcium fortification on the physicochemical properties of UHT model milk. The article was well-written and organized, and the authors clearly analyzed the properties, including pH, Zeta potential, conductivity, viscosity, particle size, and emulsion stability. This paper is a good fundamental study that can provide a guide to the addition of calcium and magnesium in the high-protein UHT beverage.

Overall response

Great thanks to the careful review and positive comments provided by reviewer #2 in our study. We would like to make the following changes regarding to the specific questions or comments from him/her.

  1. Page 6 “MgCl2 seems slightly less effective than CaCl2 in reducing Tc during heat treatment, especially in the samples with 5.0mM and 6.5mM salt addition.”  I suggest not using the word ”seem” but instead using a more solid statement since your data supports this finding.

Yes, we fully agree with reviewer #2 and removed “seems” to make the statement straightforward enough.

  1. Page 8 “This bimodality may arise due to the dual system of milk and soy protein.”  However, the author stated on page 9 that “the primary mode presented a particle size in the nano range, attributed to the oil droplets following homogenization. The secondary mode exhibited a broader range of particle sizes and was suggested to correspond to protein particles, as confirmed by microscopic examination”.  From my point of view, the two explanations for this bimodality are contradictory. Can the authors provide an explanation?

Thanks to the good comment from reviewer #2. We would like to clarify it a bit further. From our understanding, hydrolyzed soy protein has large components of insoluble proteins, and the aggregates of those insoluble particles could be responsible for the secondary peak. In the same time MPC could also have some large aggregates due to its slow or incomplete re-hydration.

In this case, the secondary peak was probably the cause of both some ISP and MPC particles together. To make it more clear, we revised the statement “This bimodality may arise due to the dual system of milk and soy protein.”  to be “This bimodality may arise due to the dual system of oil droplets coated by protein molecules on surfaces and small protein aggregates suspended after homogenization in the same system” (line 271-272)

  1. Page 8 “Based on the findings above, it can be concluded that a concentration of 3.5mM CaCl2 or MgCl2 is the maximum amount of salt allowed to prevent extensive droplet aggregation and thus still give a relatively stable milk model emulsion.” However, the author mentioned that MgCl2 is less effective than CaCl2, and thus, the max value should be different. I suggest the author use a more accurate description.

We agree with reviewer. The revised statement will be “Based on the findings above, it can be concluded that a concentration of 3.5mM for either CaCl2 or MgCl2  is still within the maximum amount of salt allowed to prevent extensive droplet aggregation and thus still give a relatively stable milk model emulsion” (line 283)

  1. Page 9 “In our study, we observed a similar shape of protein aggregates in the emulsion in spite of calcium or magnesium salts addition.” Images should be provided if you would like to discuss these results.

Thanks to the suggestion.  We would like to provide microscope image here as supplementary graphs (Fig S1, S2) to support the discussion(please see the attachment).

Round 2

Reviewer 1 Report

Comments and Suggestions for Authors

For me it is not clear, what the different letters in Table 1 mean. What is a, what is b, c, d...? The same is for Table 2. It is not explained well.

Are the errors of Tc in Table 2 realistic? How was it determined? The data quality in Figure 1 does not look so high to reach such precise results.

It is still very interesting how the suspension properties will change if both Ca and Mg salts are added. Has this been attempted or is it planned to be done?

In the conclusions, it should be supplemented with information or recommendation, up to what concentrations it would be useful to add Ca and Mg salts. Otherwise, it seems that the results obtained were already expected.

What is the role of added co-author?
